# Stability of Superhydrophobicity and Structure of PVDF Membranes Treated by Vacuum Oxygen Plasma and Organofluorosilanisation

**DOI:** 10.3390/membranes13030314

**Published:** 2023-03-09

**Authors:** Ramón Jiménez-Robles, Marta Izquierdo, Vicente Martínez-Soria, Laura Martí, Alicia Monleón, José David Badia

**Affiliations:** 1Research Group in Materials Technology and Sustainability (MATS), Department of Chemical Engineering, School of Engineering, University of Valencia, Avda. Universitat s/n, 46100 Burjassot, Spain; 2Decarbonisation Department, Plastic Technology Institute (AIMPLAS), C/Gustave Eiffel 4, 46980 Paterna, Spain; 3Department of Organic Chemistry, School of Chemistry, University of Valencia, Dr Moliner 50, 46100 Burjassot, Spain

**Keywords:** functionalisation, membrane stability, polymeric membrane, polyvinylidene fluoride, oxygen plasma, superhydrophobicity

## Abstract

Superhydrophobic poly(vinylidene fluoride) (PVDF) membranes were obtained by a surface treatment consisting of oxygen plasma activation followed by functionalisation with a mixture of silica precursor (SiP) (tetraethyl-orthosilicate [TEOS] or 3-(triethoxysilyl)-propylamine [APTES]) and a fluoroalkylsilane (1H,1H,2H,2H-perfluorooctyltriethoxysilane), and were benchmarked with coated membranes without plasma activation. The modifications acted mainly on the surface, and the bulk properties remained stable. From a statistical design of experiments on surface hydrophobicity, the type of SiP was the most relevant factor, achieving the highest water contact angles (WCA) with the use of APTES, with a maximum WCA higher than 155° for membranes activated at a plasma power discharge of 15 W during 15 min, without membrane degradation. Morphological changes were observed on the membrane surfaces treated under these plasma conditions, showing a pillar-like structure with higher surface porosity. In long-term stability tests under moderate water flux conditions, the WCA of coated membranes which were not activated by oxygen plasma decreased to approximately 120° after the first 24 h (similar to the pristine membrane), whilst the WCA of plasma-treated membranes was maintained around 130° after 160 h. Thus, plasma pre-treatment led to membranes with a superhydrophobic performance and kept a higher hydrophobicity after long-term operations.

## 1. Introduction

The interest in membrane technology is continuously increasing due to the high efficiency in separation processes, compactness of the membrane modules and lower energy consumption compared with conventional separation units that present problems related to flooding, foaming and emulsions [1,2]. In this regard, recent efforts have especially focused on the development of membrane distillation for the water desalination of seawater, brackish water and brines [3,4,5], gas separation for carbon dioxide (CO_2_) removal from flue gases and biogas [6,7], and pervaporation systems for the biobutanol and dissolved methane (CH_4_) recovery from biological effluents and wastewaters [8,9]. However, membranes tend to suffer from wetting and fouling, especially in those applications that involve highly contaminated or quite complex liquid feeds, such as industrial brines or anaerobic effluents [10,11,12,13]. Wetting and fouling phenomena lead to an additional mass transfer resistance located inside and on the surface of the membrane, respectively, reducing the separation efficiency and involving additional cleaning processes [14]. Therefore, the useful lifetime of the membrane can decline considerably, and the operational cost would rise [15,16,17]. Up to now, these issues are not completely resolved, hindering the large-scale application of membrane technology in areas such as the desalination of seawater or methane recovery from anaerobic effluents [18,19].

Different techniques for tailoring polymeric membrane characteristics have been successfully implemented to improve the wetting resistance of the membranes [20,21]. In this regard, superhydrophobic membranes with a water contact angle (WCA) higher than or equal to 150° [22] have been reported to significantly mitigate the wetting, which was attributed to the low contact area between the liquid phase and the membrane [23,24]. Additionally, superhydrophobicity has been related to the improvement in the self-cleaning properties of the membrane surface [12,19]. Thus, membrane surface functionalisation has been applied to increase its hydrophobicity at the surface level, whilst the bulk properties remain unchanged [25] by means of the addition of new hydrophobic functional groups, such as siloxanes and fluoroalkyls [26,27].

Surface modification techniques can be classified into two main categories: physical and chemical treatments [27]. Physical treatments involve a physical interaction between the modifying agents and the membrane, and the initial composition of the membrane remains unchanged. Among physical treatments, surface coatings have been widely studied to confer hydrophobicity to surfaces [28,29] and, more recently, different lithography and nanotexturing techniques have been reported to increase the fouling resistance and self-cleaning behaviour [30,31,32,33]. However, physical treatments such as surface coating could be unstable over long-term operations [34]. In contrast, chemical treatments involve the grafting of the modifying agents on the membrane surface by means of chemical bonding, such as covalent, ionic and hydrogen bonds, thus achieving a stronger adhesion force [35]. An activation step prior to the grafting of the modifying agents is frequently needed in membranes that present a high inertness and chemical stability, such as polyvinylidene fluoride (PVDF) [3,27]. This activation is usually based on the addition of oxygen-containing functional groups such as hydroxyls (-O-H), peroxides (-O-O-) and carbonyls (-C=O) that act as active sites for the subsequent grafting [36,37,38]. This activation has been successfully carried out by an alkali treatment with sodium hydroxide (NaOH) or lithium hydroxide (LiOH), plasma treatment and high-energy radiation, among other processes [3,39,40].

Plasma treatment is considered an environmentally friendly, versatile, reproducible, easily scalable and inexpensive method for activating and texturing polymer surfaces [16,41,42,43,44,45]. This technique consists of a high-energy discharge that ionises the gas near the electrodes and produces a complex gas mixture of excited ions and electrons, atoms and molecule fragments and free radicals [39,46,47]. The gas plasma is also a source of radiation that can break chemical bonds of the material [48]. The effect of the plasma discharge in the treated membrane relies on the type and conditions of the supplied gas, pressure, the power of the discharge, the duration of the treatment and the configuration of the chamber and electrodes [44,49,50,51]. Thus, a chemical and/or physical modification can be induced on the membrane surface depending on the plasma conditions [47,49] since the ion bombardment and interaction with the different reactive species contained in the plasma can produce sputtering of the membrane material (etching), substitution reactions, atom abstraction, removal of volatile substances and/or scission of polymer chains [4,46,52,53]. Additionally, the use of oxidative gases, such as oxygen (O_2_), CO_2_ and water (H_2_O), for plasma treatment creates a more reactive environment capable of adding oxygen-containing functional groups onto the membrane surface.

Plasma treatment has been reported as a useful approach for tailoring the chemical composition and/or surface morphology of different polymeric membranes and surfaces, such as polydimethylsiloxane (PDMS) [41,54,55], polypropylene (PP) [36,56], polytetrafluoroethylene (PTFE) [49], polyethylene (PE) [57], polycarbonate (PC) [33,58], polyacrylonitrile (PAN) [35,59] and polyethersulfone (PES) [35]. In contrast, PVDF has been extensively studied and commercialised as a membrane material, owing to its outstanding features such as high mechanical, chemical and thermal resistance, inertness, ease of processing and relatively low cost of the raw materials [25,27,60,61]. Hence, PVDF has been treated to improve the hydrophobicity for membrane distillation and CO_2_ absorption [24,39,62] or to induce hydrophilicity for filtration processes and enhance the fouling resistance [42,61,63]. The effects of membrane modifications with plasma treatments have mostly been evaluated based on the membrane properties at microscopic (morphology, chemical composition) and macroscopic (hydrophobicity) levels after the modification procedure, and the performance of the modified membranes is often benchmarked against the pristine membrane.

Only a few studies have evaluated the effects of long-term operation on the chemical properties, morphology and stability of hydrophobicity of the modified membranes [36]. For example, the work by Jiménez-Robles et al. [64] showed that the modification of PVDF membranes with alkali activation and functionalisation with fluoroalkylsilanes (FAS) kept a higher surface hydrophobicity and avoided the water breakthrough that non-modified PVDF suffered after approximately 800 h treating an anaerobic effluent for the dissolved CH_4_ recovery. Therefore, it is an area of interest requiring further studies into the effects of the combination of plasma activation and surface grafting on the long-term stability of membrane properties.

In this context, the aim of this work was to evaluate the effect of the oxygen plasma treatment on the surface modification of a commercial PVDF membrane. An evaluation of different silica precursors for further grafting of a FAS in order to produce superhydrophobic membranes was also carried out. First, a statistical experimental design was conducted considering the power and time of the plasma treatment and the type of silica precursors to maximise the surface hydrophobicity of the membrane by measuring the static water contact angle. Second, membrane stability tests over long-term operation were conducted in a flat-sheet membrane module with a constant flux of deionised water. The stability of the membranes was evaluated and benchmarked against the pristine PVDF in terms of hydrophobicity, thermal properties, morphology and chemical composition. 

## 2. Materials and Methods

### 2.1. Membrane Surface Functionalisation Procedure

Flat-sheet PVDF membranes (Durapore, Merck KGaA, Darmstadt, Germany) were used for the application and evaluation of different modification procedures. The membrane was composed only of hydrophobic PVDF without any support, resulting in a microporous structure. The main characteristics of the membrane are reported in Table 1.

The surface modification of PVDF was conducted with a procedure consisting of an initial activation with an oxygen plasma treatment followed by an organofluorosilanisation-based functionalisation. The activation of the membrane surface was conducted and evaluated with a vacuum plasma system (Femto, Diener electronic GmbH & Co., Ebhausen, Germany) equipped with a generator frequency of 40 kHz and a stainless-steel vacuum chamber of 100 mm diameter and 278 mm length. Initially, each membrane sample was placed in the middle of the chamber with the target surface upward, and O_2_ was flowed at 1.5 standard cubic centimetre per minute during the plasma treatment, maintaining a constant absolute pressure of 0.4 mbar inside the chamber. The power and time of the plasma treatment ranged from 3 to 17 W and 3 to 17 min, respectively, according to the design of experiments explained in Section 2.2. 

In the functionalisation step, FAS 1H,1H,2H,2H-perfluorooctyltriethoxysilane (Dynasylan^®®^ F8261, Evonik GmbH, Essen, Germany) was used as the modifying agent, and two different silica precursors (SiP), tetraethyl orthosilicate (TEOS, ≥99%, Sigma-Aldrich, USA) and 3-(triethoxysilyl)-propylamine (APTES, Sigma-Aldrich, Saint Louis, MI, USA), were evaluated. A detailed description of the functionalisation is found in earlier work [64], where the modification conditions were established based on the optimal conditions which maximised the surface WCA, using a functionalisation solution which had a FAS/SiP volumetric ratio of 0.5, a FAS/SiP concentration of 7.5%_v_ in a mixture of 2-propanol (IPA; HPLC grade, VWR Chemicals, Radnor, PA, USA) and water with a molar ratio of 57:1. After the plasma treatment, the activated membrane was immersed in the functionalisation solution for 1 h at room temperature in an orbital shaker. Afterwards, the membrane was rinsed with 2-propanol to stop the gel reaction and cured in the oven at 60 °C overnight.

For comparison purposes, coated membranes were evaluated. Through the dip-coating technique, membranes were immersed in the functionalisation solution and cured in an oven at 60 °C overnight without oxygen plasma treatment. Therefore, no chemical bonding between membranes and FAS or SiP was anticipated due to the high inertness of PVDF. The coated PVDF membranes treated with TEOS or APTES as SiP are identified as Coat-TEOS and Coat-APTES, respectively.

### 2.2. Design of Experiments and Statistical Analysis

The effect of the modification procedure with an oxygen plasma treatment on the hydrophobicity of PVDF membranes was initially evaluated on 2 cm × 2 cm membrane specimens by measuring their WCA after the functionalisation. The main variables involved in the modification procedure with the oxygen plasma treatment were analysed by means of a statistical experimental design to maximise the hydrophobicity of the PVDF membrane surface. The power and time of the plasma treatment and the SiP were the main variables (factors) that affected the membrane hydrophobicity. Table 2 shows the factor values (levels) used in the design of the experiments and statistical analysis of the functionalisation procedure. A central composite design for the analysis of variance (ANOVA) and surface response was conducted with an alpha (α) value of 1.41 and a level of confidence of 95% to identify the significant effects of the factors and their interactions on the response variable, i.e., the WCA. Then, a statistical analysis based on a multiple linear regression was conducted to determine the optimum values of the factors leading to the maximum response. The different experimental runs were conducted randomly in duplicate to avoid systematic errors. The statistical software Minitab^®®^ (Lead Technologies, Inc., Charlotte, NC, USA) was used to aid in the design of experiments and the statistical analysis.

### 2.3. Evaluation of Membrane Stability

#### 2.3.1. Long-Term Stability Tests

The tests in long-term operation with pristine, coated and plasma-activated and functionalised flat-sheet membranes were carried out to evaluate the stability of the morphological and chemical structures and hydrophobicity of the membrane under a continuous liquid flux of deionised water in the lab-scale system depicted in Appendix A. Initially, the membrane was coupled inside a 3D-printed flat-sheet module (FM) with an effective area of 25 cm^2^, and a 2-L liquid feed tank was filled with deionised water. A constant water flux was applied through the liquid side of the FM, in a closed loop, by using a peristaltic pump (Watson-Marlow Fluid Technology Solutions, Cornwall, UK). For comparison purposes and based on previous studies [64,65], the long-term tests were operated at the liquid flow rate of 21 L h^–1^ for a period of 160 h or until a constant WCA was achieved. The membrane was extracted periodically to measure its WCA in order to monitor the surface hydrophobicity as a function of time of use [28,65,66]. The morphology, thickness, chemical composition, surface porosity and thermal stability of the tested membranes were evaluated after the long-term operation.

#### 2.3.2. Thermal Analysis

Thermal analysis of the pristine, coated and plasma-activated and functionalised membranes were conducted to evaluate their thermal characteristics and crystalline nature of the PVDF structure. To determine the thermal stability of the membranes under different environments, thermogravimetric analysis (TGA) measurements were conducted in nitrogen and air atmospheres with a TGA Q5000 IR analyser (TA Instruments, New Castle, DE, USA) using high-temperature platinum pans. A 10 mg sample was placed in the holder, and the heating rate and gas flow rate were set at 10 °C min^−1^ and 50 mL min^−1^, respectively. The specimens were heated from room temperature to 800 °C. Moreover, differential scanning calorimetry (DSC) was conducted with samples of 5 mg under a flowing nitrogen atmosphere (50 mL min^−1^) at a heating rate of 10 °C min^−1^ from room temperature to 200 °C. The DSC measurements were performed with a DSC 214 Polyma (NETZSCH-Gerätebau GmbH, Selb, Germany) using aluminium pans.

The degree of crystallinity, χ (%), of the membranes was determined by using Equation (1) [38]:(1)$$\chi =\frac{{\Delta h}_{m}}{{x\Delta h}_{\alpha }+{y\Delta h}_{\beta }}\times 100\;$$
where Δh_m_ (J g^−1^) is the experimental melting enthalpy of the sample, Δh_α_ (93.07 J g^−1^) and Δh_β_ (103.40 J g^−1^) are the melting enthalpies of a 100% crystalline PVDF in the α and β phases [67], and x and y are the molar fractions of the α and β phases in the sample, respectively, which were assumed to be the predominant phases (x + y = 1). The fraction of the β phase in a sample containing both phases can be estimated using Equation (2) [68]:.
(2)$$y=\frac{A_{\beta }}{\left(\frac{K_{\beta }}{K_{\alpha }}\right)A_{\alpha }+A_{\beta }}\times 100\;$$
where A_α_ and A_β_ represent the absorbance at the wavelengths of 766 cm^−1^ and 840 cm^−1^, respectively, from the infrared spectra, and K_α_ (6.1 × 10^4^ cm^2^ mol^−1^) and K_β_ (7.7 × 10^4^ cm^2^ mol^−1^) are the absorption coefficients of the α and β phases [69]. The measured absorbance at the wavelengths of 766 cm^−1^ and 840 cm^−1^ were 0.167 and 0.187, respectively, for all the tested membranes. Measurements were taken using a Fourier transform-infrared (FTIR) spectrometer in the attenuated total reflectance (ATR) mode (Cary 630 FTIR Spectrometer, Agilent Technologies, Inc., Santa Clara, CA, USA). The fraction values of the α and β phases were 0.53 and 0.47, respectively.

### 2.4. Contact Angle Measurements

The membrane surface hydrophobicity was evaluated by means of the measurement of the static WCA following the sessile drop technique [70] by depositing a water droplet of 5.5 ± 0.1 µL onto the membrane surface using a syringe pump (KF Technology s.r.l., Italy) at room temperature. An image of the water droplet was taken at 15 s with a digital microscope (Handheld Digital Microscope Pro, Celestron LLC, Torrance, CA, USA) under white light (Philips HUE Lamp, Koninklijke Philips NV, Amsterdam, the Netherlands). The ImageJ software (National Institutes of Health, Bethesda, MD, USA) was used for image processing of the droplet profile using the contact angle plug-in based on the ellipse approximation. The WCA was evaluated at different surface locations on the membrane, and a mean value was obtained from at least four measurements using the standard deviation as a measure of the associated error. For monitoring the WCA of the membranes during the stability tests, the membranes were extracted from the FM and dried prior to the WCA measurements, removing the excess water and moisture with forced aeration at room temperature for approximately 2 h.

### 2.5. Membrane Morphology and Chemical Composition

An inspection of the membrane surface and cross-section was conducted by a field emission scanning electron microscope (FESEM) with an accelerating voltage of 10 kV (Hitachi S4800, Hitachi Ltd., Tokyo, Japan). The FESEM was equipped with an energy dispersive x-ray (EDX) spectrometer, which was used for the atomic content determination of different elements. For image acquisition, the membrane coupons were previously dried in an oven at 45 °C overnight. Afterwards, the coupons were placed on a metal holder and then coated with a fine layer of Au/Pd by sputtering in vacuum for 45 s. The surface and cross-section morphology, thickness and chemical composition were analysed. The surface porosity of the membranes was measured from the surface FESEM images by using ImageJ software [19,60,71]. The average membrane thickness and surface porosity were measured in at least six separate locations on the membrane, and the chemical composition was determined at a minimum of three locations on the surface using the standard deviation as a measure of the associated error.

## 3. Results and Discussion

### 3.1. Effect of the Oxygen Plasma Activation and Organofluorosilanisation on Membrane Hydrophobicity

A statistical experimental design evaluating the main parameters that affect the membrane surface hydrophobicity was conducted to maximise the response variable, i.e., the WCA of the plasma-activated and functionalised membranes. It is relevant to remark that after the previous plasma activation, with no further functionalisation, the droplets deposited onto the membranes were quickly absorbed, resulting in superhydrophilic surfaces with a WCA lower than 10°, as reported by other authors [38]. This hydrophilic behaviour was mainly attributed to the generation of oxygen functional groups (hydroxyls, peroxides and carbonyls) [38]. After the preliminary experiments, and based on the literature [50], the power and time of the plasma treatment and the type of SiP were found to be the main factors to be optimised. The effects of these factors and their interactions can be observed in the Pareto diagram of the design shown in Figure 1. The critical standardised effect (2.015) was calculated from the Student’s T distribution (t_α/2_,ν) with a significance level (α) of 0.05 and 16 degrees of freedom (ν), associated with the error of the design of the experiments. As observed in the Pareto diagram, the SiP presented the highest standardised effect (8.830), indicating that the WCA was mainly influenced using TEOS or APTES. Likewise, the power of the plasma treatment significantly affected the WCA in the tested conditions with a standardised effect value of 3.723. However, the plasma time showed a standardised effect of 2.045, which was similar to the critical value (2.015), indicating a low effect on the response, at least with the evaluated operational conditions. The interaction between the factors did not appear to affect the WCA of the modified membranes since lower values of the standardised effects were obtained respective to the critical value. Hence, the effects of the interactions in the WCA of the treated membranes could be neglected, thereby easing future scale-up technology.

The effects of the individual factors on the response are depicted in the main effects plot shown in Figure 2. A WCA higher than that of the pristine PVDF (119.4 ± 1.7°) was obtained for all the tested plasma-treated membranes, with an overall mean value of 153.3°, showing that the modification procedure with the oxygen plasma activation proposed in this work was suitable to obtain superhydrophobic PVDF membranes. As observed by other authors [19,72], droplets placed on the membranes for the WCA measurements easily rolled off the surface in most of plasma-treated membranes, which is an essential condition for superhydrophobic and self-cleaning surfaces.

As can be observed in the main effects plot (Figure 2), the WCA of the PVDF membranes achieved values of around 152.5° with the lowest plasma power (3 W) and maintained similar values for power less than or equal to 10 W. This could indicate that the membrane was saturated with oxygen active sites at only 3 W of power, limiting the grafting of the SiP and FAS. During plasma treatment, C-H and C-F bonds from PVDF chains are broken by the effect of the ion bombardment and radicals and electron interactions [44,51,53,67], generating volatile substances such as carbon monoxide (CO), CO_2_ or hydrogen fluoride (HF) [4,47,48] that are removed from the membrane, inducing active sites on the surface. The high reactivity of some removed substances inside the plasma chamber could lead to reabsorption on those active sites on the membrane [16]. Thus, the rate of atom removal and reabsorption on the membrane seemed to be in equilibrium under the low plasma power values of 3 to 10 W at the time points evaluated. However, the WCA of the modified membranes continuously increased with plasma power values higher than or equal to 10 W, indicating a greater performance of the plasma treatment at high power. This improvement was mainly attributed to the etching effect on the membrane surface, inducing an increase in the roughness. This phenomenon will be discussed in further sections.

The time of plasma treatment presented the lowest effect in the WCA since a slight increase in the WCA (less than 2°) was observed when increasing the plasma time from 3 to 17 min. Kim et al. [73] reported that the changes in the WCA of PVDF membranes after the oxygen plasma treatment at 10 W occurred in the first 60 s of the discharge, after which the WCA remained constant. It is worth mentioning that PVDF samples treated at a plasma power and time higher than 17 W and 17 min, respectively, became brittle and easily destroyed during their manipulation in the membrane modification and/or analysis. Hence, modified membranes could neither be evaluated nor obtained at higher plasma power and time, limiting the working upper limit of these factors for the treatment of PVDF with oxygen plasma. This is in agreement with previously reported results, which have shown an increase in membrane rigidity after the plasma treatment [74].

Regarding the main effect of the categorical factor (Figure 2), the type of SiP presented the highest effect, as previously indicated by the Pareto diagram. The highest WCA values of the plasma-treated membranes were obtained with APTES, with an overall mean value of 155°, compared to those membranes modified with TEOS, with a significantly lower overall mean value of 151°. The use of TEOS as SiP led to the seeding of siloxane chains (-Si-O-Si-O-) onto the membrane surface in the active sites generated during the plasma treatment. The reactions involved in the grafting process with TEOS are detailed elsewhere [64]. In contrast, the use of APTES can lead to the formation of siloxane chains with additional alkyl chains [39] that are naturally hydrophobic [40,50] and come from the aminopropyl group present in the APTES molecule. Moreover, the presence of an amine group can involve additional grafting reactions in the oxygen-rich active sites on the plasma-treated surfaces. Such reactions include amide formation and even breakage or scission of the PVDF backbones [22,26,36,75]. Hence, the incorporation of an additional reactive group, such as the primary amine of the APTES, can lead to the formation of a more complex and uneven structure on the membrane surface, compared to the more symmetric siloxane structure originated from TEOS. Likewise, subsequent grafting of the FAS could have taken place in a heterogeneous way when APTES was used, which positively increased the WCA.

An ANOVA and surface response analysis based on a linear multiple regression were conducted to create a model that fit the experimental results and to determine the values of the factors that maximised the WCA of the plasma-treated membranes. The response surface for each categorical factor value (TEOS and APTES) is shown in Figure 3. From the ANOVA of the model, the F-values of the quadratic and two-way interaction terms were low (less than 2), indicating that they could be neglected compared to the lineal terms with a higher F-value of 32, which agreed with the previous main effects analysis. In addition, the ANOVA showed no evidence of lack-of-fit (*p*-value of 0.58), indicating that the model can adequately predict the WCA of the modified membranes under the operational conditions tested, as observed when comparing the response surface with the experimental data (open symbols in Figure 3). The maximum WCA values predicted by the model were 155.2 ± 2.7° and 157.3 ± 2.1° with TEOS and APTES, respectively, at the highest plasma power and time of 17 W and 17 min, respectively. However, these maximum WCAs did not show significant differences compared with those WCAs predicted at a lower plasma power and time of 15 W and 15 min, respectively, and the experimental WCAs obtained at these conditions were 155.5 ± 1.5° and 157.0 ± 0.9° for TEOS and APTES, respectively. Thus, the conditions for the plasma treatment were established at 15 W and 15 min in order to achieve high superhydrophobic membranes for their further characterisation. The oxygen plasma-activated and functionalised PVDF membranes at these conditions and TEOS or APTES as SiP are labelled as PO_2_-TEOS and PO_2_-APTES, respectively. The infrared spectra of these membranes and the pristine PVDF are shown in Appendix A, even though no significant differences were observed due to the penetration depth of the FTIR-ATR analysis [21].

Only a few works evaluating a membrane modification procedure with plasma activation followed by grafting of FASs can be found in the literature. In a similar approach to this work, Liu et al. [19] activated a PVDF membrane with oxygen plasma at 50 W for 1 min, followed by grafting of the FAS 1H,1H,2H,2H-perfluorodecyltriethoxysilane, achieving a WCA of 162.0 ± 2.3° in membrane surfaces with a pillar surface structure. The higher WCA reported in that work could be ascribed to the higher fluorine chain length of the modifying agent used [39] and the high initial WCA of the pristine membrane (155.3 ± 1.7°) [19]. Sairiam et al. [39] also evaluated the previous FAS for the modification of PVDF membranes with a helium plasma activation at 80 W for 180 s, and they reported an increase in the WCA from 68.9 ± 0.9° to 145.6 ± 3.1°.

### 3.2. Membrane Stability Tests in Long-Term Operation

Long-term performance is a design requirement of high relevance for polymer-based dispositive [76], but research works evaluating the effect of long-term operation on the stability of modified membranes are still very scarce and mainly focused on the stability in the separation performance [39].

In this section, the effect of long-term operation on the stability of different membrane properties was evaluated. First, the results regarding the stability of the membrane bulk properties after long-term operation are shown, followed by the stability evaluation of the surface properties during the operation. For comparison purposes, the outcomes obtained with the different treated membranes were benchmarked with the pristine PVDF membrane (p-PVDF).

#### 3.2.1. Stability of the Bulk Properties after Operation

Thermal analysis is a quality tool to characterise the performance of polymers at the design stage and after operation [77]. The results from thermogravimetric analysis under different atmospheres are shown in Figure 4 in the form of derivative thermogravimetric (DTG) curves for the pristine, coated and plasma-treated PVDF membranes before and after the long-term stability tests. Under an oxidative environment, all the membranes before the stability test showed a similar trend until the complete decomposition of the samples was observed in the range of 450 to 600 °C (Figure 4a). Under an inert environment (Figure 4c), the decomposition profile was similar to the outcomes under an oxidative atmosphere, showing main decomposition peaks at temperatures around 475 °C. However, a char was observed at the end of the TGA analysis for all the PVDF membranes representing 20 to 30% of the initial weight of the sample, similar to the pristine PVDF analysed by other authors [59]. These outcomes indicate that the chemical structure of the pristine PVDF was not affected by plasma activation and functionalisation.

The DTG thermograms of the membranes after long-term operation (Figure 4b,d) showed the same trends that the non-used membranes had with no significant variation in the decomposition temperatures, indicating that long-term operation did not affect the thermal behaviour and stability of the bulk membrane.

DSC measurements were conducted to detect potential modifications in the amorphous/crystalline configuration and thermal performance of the membranes after the modification procedures and operation. The DSC thermograms of the pristine, coated and modified PVDF membranes under a nitrogen atmosphere are shown in Figure 5. The melting temperature and enthalpy, together with the crystallinity degree before and after the long-term stability tests, are shown in Table 3. Similar values were observed for all membranes, regardless of the membrane treatment and operation, which highlights the stability of the bulk PVDF toward surface modification technologies, as stated by other authors [35,39,43,48]. The characteristic melting temperature was approximately between 160 and 180 °C, with an endothermic peak at around 163 °C. These outcomes are consistent with the literature [38,67].

The melting enthalpies of the samples were between 53 and 59 J g^−1^, and maximum crystallinity with a value of 61% was observed for the pristine PVDF. The treated membranes showed slightly lower crystallinity degrees with values between 54 and 58%. Other authors have also reported similar crystallinity degrees of around 54% for PVDF membranes treated with argon and oxygen plasma at 100 W for a period ranging from 200 to 600 s [38].

Similar findings and trends were observed on the membranes after the stability tests (Figure 5b and Table 3). In general terms, neither membrane treatment nor long-term operation induced any significant modification of the thermal behaviour of the membranes. The works of Correia et al. [38,67] also reported no significant effect of argon and oxygen plasmas on the crystallinity of PVDF membranes.

#### 3.2.2. Stability of the Surface Hydrophobicity in Operation

The stability of PVDF membranes during long-term operation was evaluated by monitoring its WCA as a useful indicator of changes or alterations occurring on the membrane surface [28,64,65,66]. The results of the variation in WCA with the time of use of the coated PVDF membranes (Coat-TEOS and Coat-APTES) and plasma-treated PVDF membranes at the optimal conditions (PO_2_-TEOS and PO_2_-APTES) are shown in Figure 6a,b, respectively. The initial WCA of Coat-TEOS and Coat-APTES were 144.8 ± 3.2° and 149.8 ± 2.7°, respectively. These values were lower than their respective plasma-treated membranes (PO_2_-TEOS and PO_2_-APTES), suggesting a lower presence of FAS and SiP on the membrane surface due to the absence of a previous activation. Kaur et al. [25] studied the grafting of methacrylic acid on non-treated and plasma-treated PVDF membranes, and they reported that grafting was facilitated by the formation of radicals and peroxide groups in argon plasma-treated membranes.

As can be seen in Figure 6a, the WCA of both coated membranes decreased pronouncedly during the first 24 h, reaching similar values to those of the p-PVDF. Then, the WCA of the coated membranes remained constant after 24 h, as did the p-PVDF. Hence, it can be concluded that the effect of the coating with FAS/TEOS and FAS/APTES solutions was completely lost from the surface-coated membranes before 24 h. This fact suggests that only a physical deposition of FAS and SiP was involved without further chemical reactions, creating an unstable surface layer that would be removed by dragging effects under a constant liquid flux over the membrane [34,64].

In contrast, although the WCA of the plasma-treated membranes PO_2_-TEOS and PO_2_-APTES decreased from the initial values of 155.5 ± 1.5° and 157.0 ± 0.9° to 127.0 ± 3.4° and 129.9 ± 3.8° at approximately 75 h, respectively, the WCA values stayed almost constant and slightly higher than those of the p-PVDF after 75 h (Figure 6b). These results suggested that the effect of the functionalisation was partially lost, likely due to removal of grafted molecules by the dragging effect, as reported in a previous work with modified membranes activated with an alkali solution [64]. However, WCA of the plasma-treated membranes after the stability tests was slightly higher than that of p-PVDF, suggesting a remnant of FAS and/or SiP on the surface. Hence, stronger chemical bonding and interactions could be inferred between the SiP, FAS and activated PVDF than those shown by the coated membranes. Comparing the effect of the SiP on the membrane stability, both PO_2_-APTES and PO_2_-TEOS membranes experienced a similar decrease rate with the time of use. However, PO_2_-APTES always showed slightly higher WCAs, mainly attributed to the additional hydrophobic aminopropyl segments from the APTES grafted on the membrane, as previously discussed. 

Research works evaluating the effect of long-term operation on the stability of modified membranes are still very scarce, with most of them focusing on the stability of the separation performance. For example, our previous study [64] evaluated a functionalisation protocol using an alkali-based activation and showed that the WCA of the modified PVDF decreased to values lower than the pristine membrane in the first 50 h of operation at a liquid flow rate of 27 L h^−1^. In this regard, the plasma pretreatment seemed to provide a more stable hydrophobic surface. Moreover, Sairiam et al. [38] performed long-term tests for CO_2_ absorption with helium plasma-treated and functionalised PVDF membranes, and they reported a stable gas flux during the 15 days, unlike the unmodified membrane and the membrane activated with an alkali solution. Liang et al. [60] evaluated the stability of PVDF membranes modified with Ar plasma activation and grafting with methacrylic acid by immersing the membranes in solutions with different pH for only 15 min, and they reported a similar WCA before and after the stability tests. Moreover, Gryta [36] reported that PP membranes treated with helium (He) plasma for enhancing hydrophilicity showed a greater and more stable performance than non-treated membranes during 300 h of treating actual seawater for water purification, in which a higher water flux and lower permeate conductivity have been reported.

### 3.3. Structure and Chemical Composition of the Modified Membranes

Microscopy using FESEM and EDX analyses was conducted in order to determine the morphology of the surface, cross-section and chemical composition of the pristine, coated and plasma-treated membranes, both before and after the stability tests. The FESEM images of the surface and cross-section of the different analysed membranes before the stability tests are shown in Figure 7. The surface morphology of the Coat-TEOS and Coat-APTES membranes (Figure 7b1,c1) was similar to that of the p-PVDF (Figure 7a1), and a surface coating layer covering the membrane surface was not observed. In addition, the measured surface porosities of the pristine and coated membranes were similar, with values around 10% (Table 4).

The surface morphology of the plasma-treated membranes (Figure 7d1,e1) was similar to that of the pristine PVDF when a low plasma power and time of 5 W and 5 min were applied, respectively, independently of using TEOS or APTES as SiP. In contrast, at a higher plasma power and time of 15 W and 15 min, respectively, the surface became a pillar-like structure, resulting in a rougher and more open surface structure with larger pore sizes (Figure 7d1,e1), and the surface porosity increased to approximately 15 % (Table 4) for both PO_2_-TEOS and PO_2_-APTES. This increase in the surface roughness also contributed to the increase of the WCAs, leading to superhydrophobic membranes in which the droplets easily rolled off, suggesting a Cassie state, which is desirable for wetting and fouling resistant surfaces [33,55]. This severe change at the surface was explained by the etching effect of plasma treatment, especially at a plasma time higher than or equal to 15 min [46].

Different authors have reported similar observations regarding surface morphology changes after plasma treatment. Lin et al. [43] reported an increase of the pores on the surface caused by etching during a plasma treatment with methane at a power higher than 50 W and time less than 5 min, and they suggested that the formation and breakage rates of C-F bonds were equal for plasma time periods higher than 5 min. Liang et al. [61] also reported an increase of the surface pore size and porosity of PVDF membranes after argon plasma treatment at 18 W and time less than or equal to 120 s, attributed to the etching effect. Yang et al. [62] obtained a more open structure on PVDF membranes after plasma treatment with carbon tetrafluoride at 150 W and 15 min. Xu et al. [60] observed an increase in the surface porosity from 13 to 34% after an argon plasma treatment at 30 W and 120 s, and this value declined to 12% after the grafting of an organosilane similar to the FAS used in this work. Jeong et al. [4] observed that the PVDF surface became a pillar-like structure after an oxygen and carbon tetrafluoride plasma treatment at 62 W for plasma time periods higher than 30 min.

The FESEM images of the membrane cross-section before the stability tests were taken and shown in Figure 7. They focused on the target surface at high magnification. In Appendix A, images were taken at lower magnification to show the entire membrane. The cross-section of the upper layer of p-PVDF (Figure 7a2) changed after the coating treatment (Figure 7b2,c2), showing a more porous structure. Only in the case of the plasma-treated membranes under soft conditions of 5 W and 5 min (Figure 7d2,e2), a dense-like cross-section was observed at the upper surface (≤7 µm), likely owing to the grafting of SiP and FAS, especially when TEOS was used, as observed in our previous study [64]. In contrast, the upper layers of the plasma-treated membranes at 15 W and 15 min showed a highly porous structure and rougher surface (Figure 7d2,e2), in which a surface profile with ridges and valleys was observed, as reported by other authors [38]. Regarding the thickness of the PVDF membranes, it remained unchanged in values ranging from 120 to 130 µm after the coating treatment and plasma activation and functionalisation (Appendix A), as reported by other authors [24].

After the stability tests, all the membranes experienced a reduction in surface porosity to a value of approximately 5% (Table 4), which was attributed to plastic deformation observed at the surface level (Figure 8), which also contributed to the WCA decrease during the stability test. However, no significant changes were observed on the cross-section with membrane thickness values of approximately 120 µm (Figure 8). This indicated a high membrane bulk stability and mechanical resistance under the tested operational conditions. 

The surface chemical composition of the pristine, coated and plasma-treated membranes was determined by the EDX, and the atomic ratios of F, O, Si and N with respect to C (%_atomic_/%_atomic_) are shown in Table 5. The F/C ratio of p-PVDF was 0.92, quite far from the theoretically value of 1.00 for pure PVDF, indicating a raw PVDF membrane with the presence of other organic carbon compounds.

After the coating of PVDF, in both Coat-TEOS and Coat-APTES membranes, the F/C ratio slightly increased to 0.95 due to the deposition of FAS. The presence of O, Si and N indicated the deposition of SiPs. A similar deposition grade of FAS and SiP could be inferred independently of the SiP applied since similar atomic ratios were obtained. The highest F/C ratio of 0.99 was obtained for the PO_2_-APTES membrane, indicating a higher grafting grade of FAS with respect to the coated membranes and PO_2_-TEOS. This can help explain the greater hydrophobicity of the membranes treated with oxygen plasma and APTES. Both PO_2_-TEOS and PO_2_-APTES membranes presented slightly higher O/C, Si/C and N/C ratios than Coat-TEOS and Coat-APTES, respectively, indicating a higher presence of SiP molecules because of the generation of active sites on the membrane during the plasma treatment. The higher O/C ratios in the plasma-treated membranes with respect to the coated membranes could also be attributed to the increase in the oxygen content during the plasma treatment [48,53,67].

After the stability tests, the F/C ratio decline was more noticeable in the plasma-treated and functionalised membranes (Table 5), indicating a loss of fluorine in accordance with the decrease of the WCA with the time of use. The F/C ratio decreased from 0.95 to 0.80 and 0.99 to 0.85 in the PO_2_-TEOS and PO_2_-APTES membranes, respectively. These outcomes showed a similar decrease of around 15% in the F/C ratio, suggesting that a similar amount of FAS moieties was removed. In addition, the O/C, Si/C and N/C ratios stayed the same or slightly higher after the stability test, which could indicate that the condensation and grafting of SiP led to a stronger interaction/chemical bonding with the membrane than those involved with FAS, hindering the dragging of SiP by the liquid flux. Furthermore, the higher amount of fluorine and SiP on the PO_2_-APTES after the stability tests could explain its higher WCA with respect to the PO_2_-TEOS at the end of the test. The F/C ratio of the coated membranes decreased slightly after the stability test and kept at similar values to that of the pristine membrane, indicating a loss of the coating layer and showing similar WCAs. In addition, the F/C ratio of the plasma-treated membranes was lower than those of the pristine and coated membranes despite their higher WCAs after the stability tests. This result suggests that the higher surface roughness and pore size had a predominant effect on the WCA.

From the previous results, new modification protocols should be evaluated in future studies in this direction, with special focus on the stability of grafting compounds on the modified membranes, preventing the removal or dragging of the modifying agents under moderate liquid fluxes at long-term operations. This prolonged useful lifetime of the functionalisation layer would lead to a reduction in the operational costs during large-scale applications.

## 4. Conclusions

Superhydrophobic PVDF membranes were prepared by means of surface treatment based on an initial oxygen plasma activation followed by functionalisation with a fluoroalkylsilane and silica precursors. The outcomes of the statistical design of the experiments showed that the type of silica precursor had the highest effect on the water contact angle of the modified membrane. Particularly, the use of APTES as silica precursors always led to membranes with higher water contact angles than those obtained by using TEOS, which was ascribed to the asymmetric structure of APTES. Other factors, such as the power and, especially, the time of the plasma discharge, showed less significant effects on the water contact angle. From those results, optimal values for the oxygen plasma treatment were established at a power of 15 W and a time of 15 min, obtaining contact angles higher than 155° and avoiding membrane degradation. In fact, the modifications acted primarily on the surface, and the bulk properties remained stable and maintained the thermal features of PVDF membranes after the treatments. Moreover, at those plasma conditions, the membrane surface showed a pillar-like morphology, with higher porosity and roughness, which favoured the hydrophobicity.

During long-term operation tests under a continuous liquid flux, the contact angle decreased with the time of use, which was attributed mainly to the removal of fluoroalkylsilane molecules from the surface since a lower fluorine content was observed after the long-term tests, and also to the reduction of the surface porosity due to a surface plastic deformation. However, the functionalisation layer of the oxygen plasma-activated membranes showed high stability, in comparison with the coated membranes without any activation step. After 160 h of operation, the plasma-treated membranes showed a water contact angle of approximately 130°, which was still higher than that of the non-modified membranes (119°). Further research should focus on the improvement of the anchoring of the modifying agents onto the PVDF substrate and the feasibility and scalability of the functionalisation methodologies for their implementation at an industrial scale.

## Figures and Tables

**Figure 1 membranes-13-00314-f001:**
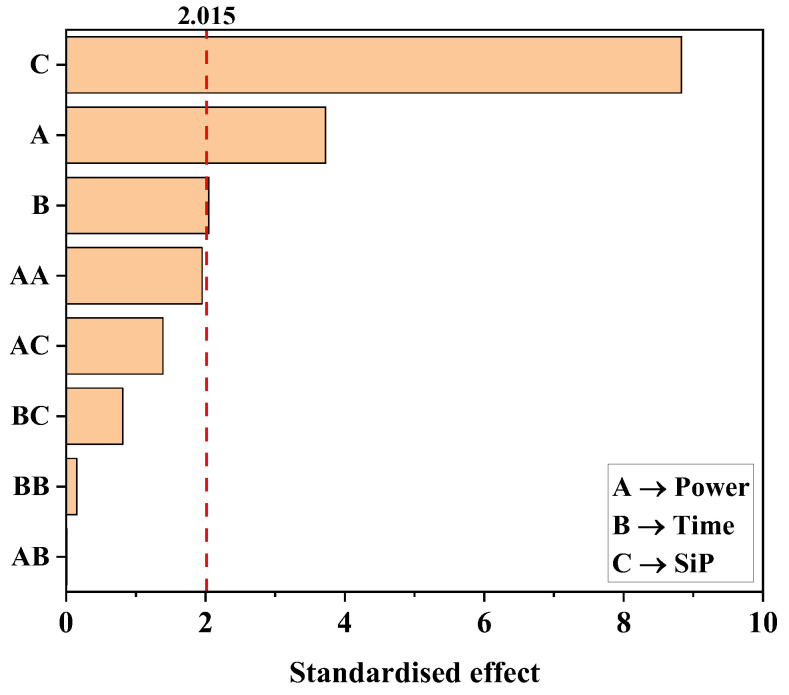
Pareto diagram of the standardised effects of the factors analysed in the central composite design (power and time of the oxygen plasma and silica precursor [SiP]). The dashed red line represents the critical standardised effect for a level of confidence of 95%.

**Figure 2 membranes-13-00314-f002:**
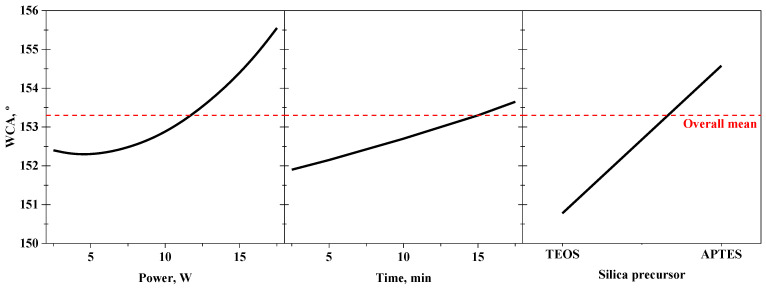
Main effects plot of the individual factors analysed in the central composite design (power and time of the oxygen plasma and silica precursor).

**Figure 3 membranes-13-00314-f003:**
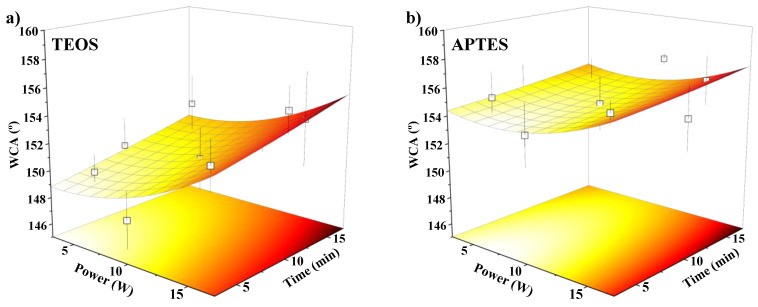
Response surface obtained from the central composite design for (**a**) TEOS and (**b**) APTES as silica precursors compared to experimental data (open square—□).

**Figure 4 membranes-13-00314-f004:**
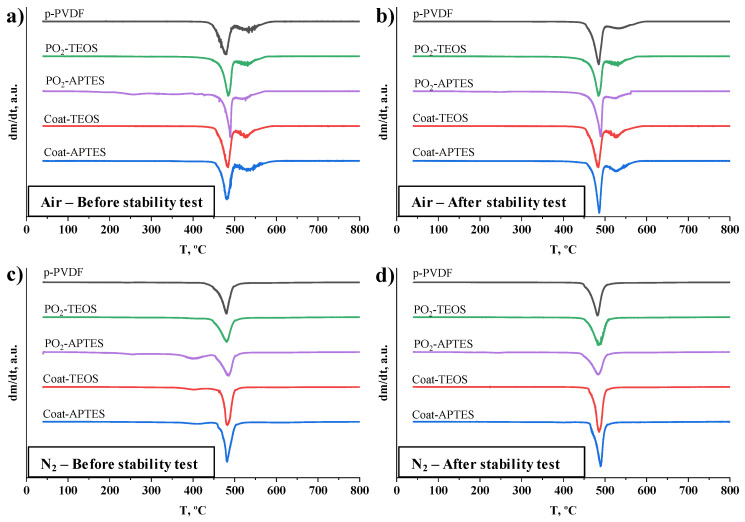
Thermogravimetric analyses under an oxidative atmosphere with air (**a**,**b**) and an inert atmosphere with nitrogen (**c**,**d**) of the pristine PVDF (p-PVDF) membrane, coated membranes with FAS/TEOS and FAS/APTES (Coat-TEOS and Coat-APTES, respectively) and modified membranes with oxygen plasma activation at 15 W for a period of 15 min and functionalised with FAS/TEOS and FAS/APTES (PO_2_-TEOS and PO_2_-APTES) before (**a**,**c**) and after (**b**,**d**) long-term stability tests. The derivative thermogravimetric curves (dm/dt) as the derivative of the mass (%) with respect to the time (min) as a function of temperature (°C) and expressed in arbitrary units (a.u.).

**Figure 5 membranes-13-00314-f005:**
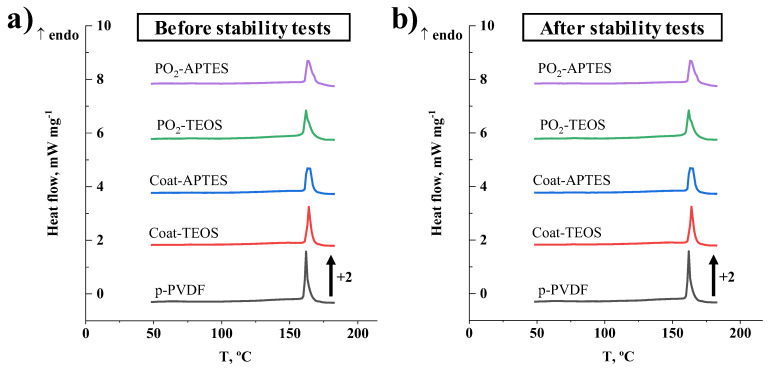
Differential scanning calorimetry of the pristine PVDF membrane (p-PVDF), coated membranes with FAS/TEOS and FAS/APTES (Coat-TEOS and Coat-APTES, respectively) and modified membranes with oxygen plasma activation at 15 W for a period of 15 min and functionalised with FAS/TEOS and FAS/APTES (PO_2_-TEOS and PO_2_-APTES, respectively) (**a**) before and (**b**) after the stability tests.

**Figure 6 membranes-13-00314-f006:**
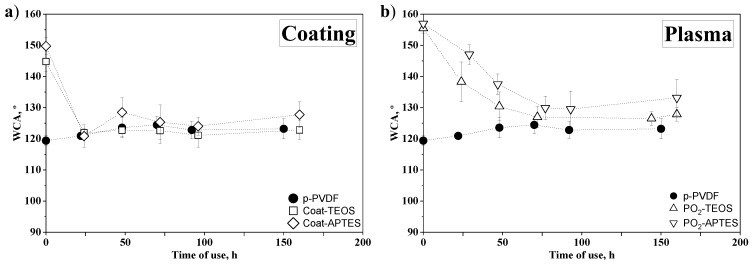
(**a**) Water contact angle versus time of use for pristine PVDF membrane (p-PVDF) and coated membranes with FAS/TEOS and FAS/APTES (Coat-TEOS and Coat-APTES, respectively) and (**b**) p-PVDF and modified membranes with oxygen plasma activation at 15 W for a period of 15 min and functionalised with FAS/TEOS and FAS/APTES (PO_2_-TEOS and PO_2_-APTES, respectively).

**Figure 7 membranes-13-00314-f007:**
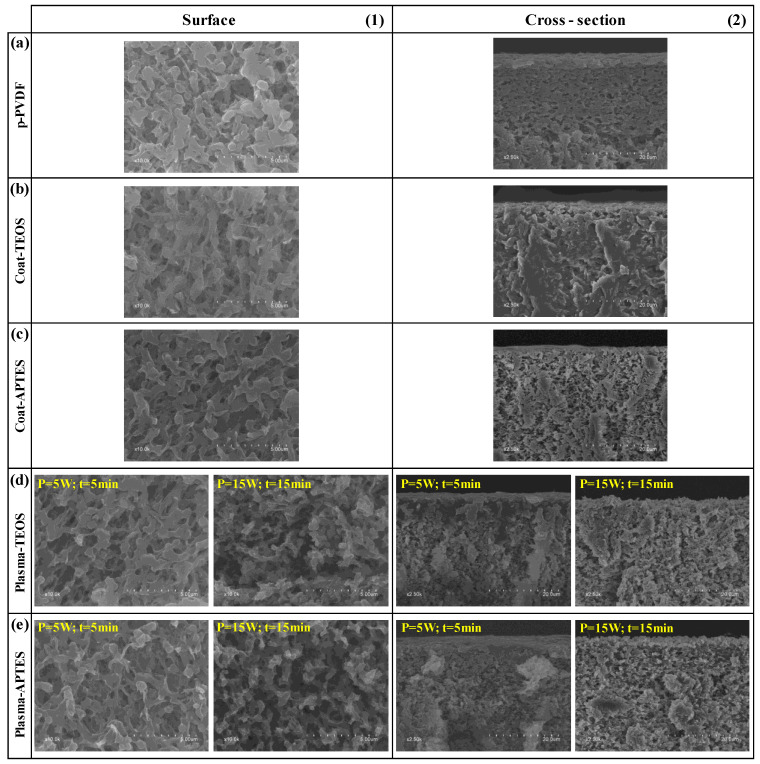
FESEM images of the (**1**) surface and (**2**) cross-section of the (**a**) pristine PVDF (p-PVDF), coated membranes by (**b**) FAS/TEOS (Coat-TEOS) and (**c**) FAS/APTES (Coat-APTES) and modified membranes with oxygen plasma activation at different plasma conditions and functionalised with (**d**) FAS/TEOS (Plasma-TEOS) and (**e**) FAS/APTES (Plasma-APTES).

**Figure 8 membranes-13-00314-f008:**
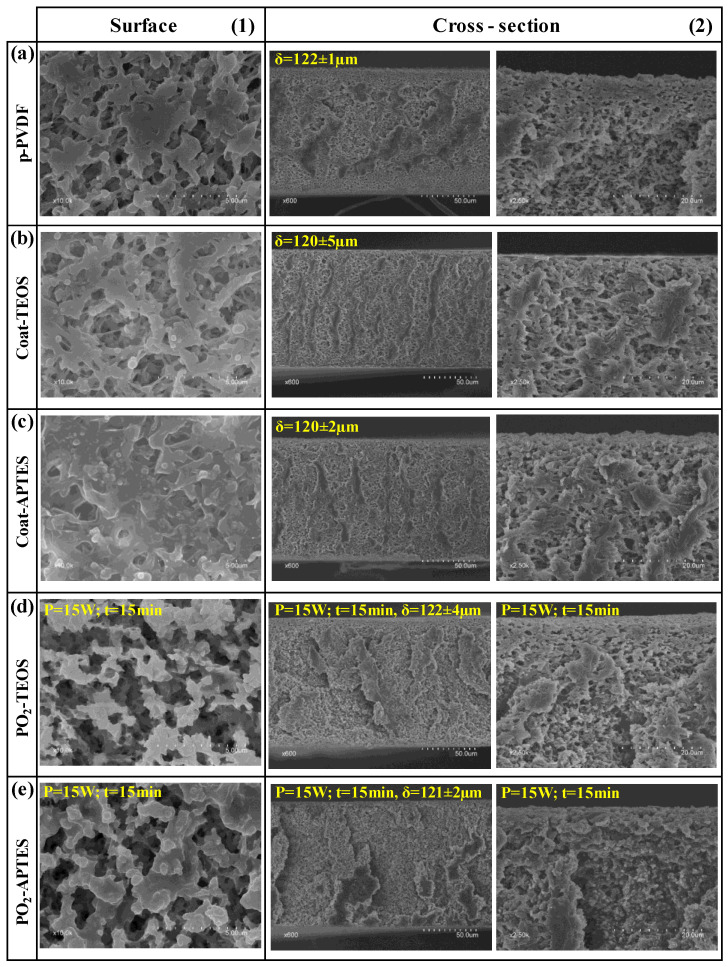
FESEM images of the (**1**) surface and (**2**) cross-section of the (**a**) pristine PVDF (p-PVDF), coated membranes by (**b**) FAS-TEOS (Coat-TEOS) and (**c**) FAS-APTES (Coat-APTES) and modified membranes with oxygen plasma activation at 15W and 15 min and functionalised with (**d**) FAS/TEOS (PO_2_-TEOS) and (**e**) FAS/APTES (PO_2_-APTES) after the long-term stability test.

**Table 1 membranes-13-00314-t001:** Characteristics of the flat-sheet PVDF membrane.

Property	Value
Structure	Microporous
Thickness, µm	125 ^a^
Pore diameter, µm	0.22 ^a^
Bubble point, bar	≥1.24 ^a^
Porosity, %	75 ^a^
Static water contact angle, °	119.4 ± 1.7 ^b^

^a^ Provided by the Merck KGaA. ^b^ Measured as explained in Section 2.4.

**Table 2 membranes-13-00314-t002:** Factors and levels used in the design of the experiments and statistical analysis (with α = 1.41) of the functionalisation procedure.

Independent Variables (Factors)		Levels				
		−α	−1	0	+1	+α
A	Power (W)	3	5	10	15	17
B	Time (min)	3	5	10	15	17
C	Silica precursor (SiP) *	TEOS—APTES				

* Categorical factor.

**Table 3 membranes-13-00314-t003:** Effect of different modification methods before and after the stability tests on the melting temperature (T_m_), enthalpy (Δh_m_) and the degree of crystallinity (χ) in the pristine PVDF membrane (p-PVDF), coated membranes with FAS/TEOS and FAS/APTES (Coat-TEOS and Coat-APTES, respectively) and modified membranes with oxygen plasma activation at 15 W for a period of 15 min and functionalised with FAS/TEOS and FAS/APTES (PO_2_-TEOS and PO_2_-APTES, respectively).

	Before Stability Test			After Stability Test		
	T_m_, °C	Δh_m_, J g^−1^	χ, %	T_m_, °C	Δh_m_, J g^−1^	χ, %
p-PVDF	162.3	59.3	60.6	162.0	62.2	63.5
Coat-TEOS	164.0	53.0	54.1	164.0	54.9	56.1
Coat-APTES	164.3	55.0	56.2	162.6	54.7	55.9
PO_2_-TEOS	162.3	54.2	55.4	161.8	58.1	59.3
PO_2_-APTES	162.9	57.0	58.2	163.4	58.9	60.2

**Table 4 membranes-13-00314-t004:** Surface porosity (%) of the pristine PVDF (p-PVDF), coated PVDF membranes with FAS/TEOS (Coat-TEOS) and FAS/APTES (Coat-APTES) and modified membranes with oxygen plasma activation at 15 W for a period of 15 min and functionalised with FAS/TEOS (PO_2_-TEOS) and FAS/APTES (PO_2_-APTES) before and after its use in the long-term stability tests.

Membrane	Before Stability Test	After Stability Test
p-PVDF	11 ± 3%	5 ± 1%
Coat-TEOS	8 ± 1%	4 ± 2%
Coat-APTES	10 ± 2%	3 ± 3%
PO_2_-TEOS	15 ± 4%	6 ± 3%
PO_2_-APTES	15 ± 3%	7 ± 1%

**Table 5 membranes-13-00314-t005:** Fluorine-, Oxygen-, Silicon- and Nitrogen-to-Carbon atomic ratio of the pristine PVDF membrane (p-PVDF), coated membranes with FAS/TEOS (Coat-TEOS) and FAS/APTES (Coat-APTES) and modified membranes with oxygen plasma activation at 15 W and 15 min and functionalised with FAS/TEOS (PO_2_-TEOS) and FAS/APTES (PO_2_-APTES) before and after its use in the long-term stability tests. Standard deviation ≤ 0.05.

	Before Stability Test				After Stability Test			
Membrane	F/C	O/C	Si/C	N/C	F/C	O/C	Si/C	N/C
p-PVDF	0.92	-	-	-	0.90	-	-	-
Coat-TEOS	0.95	0.02	<0.01	-	0.94	0.02	<0.01	-
Coat-APTES	0.95	0.03	<0.01	0.03	0.91	0.04	0.01	<0.01
PO_2_-TEOS	0.95	0.04	0.01	-	0.80	0.03	<0.01	-
PO_2_-APTES	0.99	0.05	0.01	0.04	0.85	0.09	0.02	0.04

## Data Availability

Not applicable.

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
