# Peer review of "Stability of Superhydrophobicity and Structure of PVDF Membranes Treated by Vacuum Oxygen Plasma and Organofluorosilanisation"

_membranes, 2023, doi:10.3390/membranes13030314_

Round 1

Reviewer 1 Report

The authors reported the stable Superhydrophobic poly(vinylidene fluoride) (PVDF) membranes obtained by a surface treatment of oxygen plasma followed by functionalization with a mixture of silica precursor (SiP) (tetraethyl-orthosilicate [TEOS] or 3-(triethoxysilyl)-propylamine [APTES]) and a fluoroalkylsilane (1H,1H,2H,2H-perfluorooctyltriethoxysilane). 

The work may be published after a major revision. 

1. The authors were positive about the result after the plasma treatment, however, in my view, it is quite disappointing - the contact angle didn't increase much and the contract angle still dropped significantly after a continuous aqueous flex. The author should also comment on the limitations and drawbacks comparing to other methods and other materials in the conclusion section. Especially, the result clearly supports that PVDF doesn't react with plasma effectively. It is known that other C-H based polymer materials will be readily activated by plasma and may result in a more stable hydrophobic surface after the silane treatment. The comparison between C-H and C-F materials should be included. 

2. What are the OR groups on PVDF after the plasma treatment? If no data on this, at least some references should be added. 

Reviewer 2 Report

The paper is quite interesting. For clarity, I suggest to the authors to add some more explanations/data about these points:

- FT-IR or Raman of the membranes should be added to control the effect on the surface chemistry of the O2 plasma treatments and of the functionalization with the different SiP . These measurements could confirm the sentences in lines 339-350

- Did the author check the thickness of the Si-coatings? This could be an important parameter in comparison of the long-term properties and in the thermal behviour

Round 2

Reviewer 1 Report

ok to accept the current form, although teh authors didn't fully address my comments. My original comment was on why perfluorocarbon polymer is used since CH based polymer showed a better effect. 

Reviewer 2 Report

The authors replied to the comments. The paper is suitable for publication